# Effect of Fermentation Humidity on Quality of Congou Black Tea

**DOI:** 10.3390/foods12081726

**Published:** 2023-04-20

**Authors:** Sirui Zhang, Xinfeng Jiang, Chen Li, Li Qiu, Yuqiong Chen, Zhi Yu, Dejiang Ni

**Affiliations:** 1National Key Laboratory for Germplasm Innovation and Utilization of Horticultural Crops, Wuhan 430070, China; 2College of Horticulture and Forestry Sciences, Huazhong Agricultural University, Wuhan 430070, China; 3Jiangxi Sericulture and Tea Research Institute, Nanchang 330202, China

**Keywords:** congou black tea, fermentation, humidity, quality, bioactivity

## Abstract

This study investigated the effect of different fermentation humidities (55%, 65%, 75%, 85% and 95%) on congou black tea quality and bioactivity. Fermentation humidity mainly affected the tea′s appearance, aroma and taste quality. The tea fermented at low humidity (75% or below) showed a decrease in tightness, evenness and moistening degree, as well as a heavy grassy and greenish scent, plus a green, astringent and bitter taste. The tea fermented at a high humidity (85% or above) presented a sweet and pure aroma, as well as a mellow taste, plus an increase of sweetness and umami. With increasing fermentation humidity, the tea exhibited a drop in the content of flavones, tea polyphenols, catechins (EGCG, ECG) and theaflavins (TF, TF-3-G), contrasted by a rise in the content of soluble sugars, thearubigins and theabrownins, contributing to the development of a sweet and mellow taste. Additionally, the tea showed a gradual increase in the total amount of volatile compounds and in the content of alcohols, alkanes, alkenes, aldehydes, ketones and acids. Moreover, the tea fermented at a low humidity had stronger antioxidant activity against 2, 2-Diphenyl-1-picrylhydrazyl (DPPH) and a higher inhibiting capability on the activities of α-amylase and α-glucosidase. Overall results indicated the desirable fermentation humidity of congou black tea should be 85% or above.

## 1. Introduction

Black tea is a leading consumer tea worldwide, with antioxidant, anti-hypertension, hypoglycemic and cardiovascular disease prevention functions [1]. The black tea manufacturing process mainly includes the steps of withering, rolling (rolling-cutting), fermentation and drying, with fermentation being a very important step due to the formation of the main taste substances and the accumulation of aroma substances at this stage [2]. Under the impact of polyphenol oxidase and peroxidase, enzymatic reactions occur in polyphenolic components in black tea during fermentation, leading to their transformation into a variety of characteristic oxidation products, such as tea pigments, which contribute to the generation of the unique aroma, taste, color and function of black tea [3,4].

The primary conditions influencing the fermentation of black tea are temperature, oxygen content, time, humidity, etc. When comparing black tea fermented at different temperatures, fermentation at 30 °C to 32 °C was beneficial to improving the sensory quality of black tea, and fermentation of black tea at 25 °C to 28 °C contributed to improving its antioxidant capacity, inhibiting the activities of α-amylase and α-glucosidase, and suppressing intestinal glucose absorption [5]. Chen et al. demonstrated that, compared to low and normal oxygen environments during fermentation, an adequate oxygen environment could stimulate the oxidation of flavonoid glycosides, catechins and certain phenolic acids, as well as glutamic acid accumulation, thus reducing black tea bitterness and astringency while increasing its umami [6]. Wang et al. found that for Yunnan congou black tea, 3 h of fermentation time favored the preservation of simple catechins, the conversion of bitter and astringent substances, the formation of soluble sugars and the accumulation of aroma compounds [7]. Hua et al. illustrated that fermentation temperature (25 °C to 30 °C) and fermentation time (60 to 90 min) were the crucial conditions affecting the conversion rate of phenolic compounds and fermentation direction. The brightness of tea liquid decreased with the increase of fermentation time, and fermentation at appropriate low temperatures contributed to the retention of polyphenol oxidase activity and the generation of good liquid color [8]. Fermentation humidity generally refers to the relative humidity in the environment during tea fermentation, i.e., the degree of water vapor in the wet air apart from saturation. Moisture is a direct participant and an essential solvent medium for the chemical reactions occurring in the fermentation of black tea. Changes in the relative humidity of the fermentation environment can affect the evaporation of water from the fermented leaves, thus changing their moisture content and affecting the chemical changes plus the type and quantity of products formed in the fermentation process [9]. At present, except for the limited research on the fermentation humidity gradient, few systematic studies have been devoted to black tea fermentation humidity at home and abroad [10]. In actual tea production, the fermentation humidity is usually determined based on experience, lacking a scientific and theoretical basis.

Black tea can be divided into two categories based on their shape: congou black tea and broken black tea. The difference between the two in process is mainly determined by the rolling and rolling-cutting methods, and the resulting changes during fermentation. After withering, congou black tea is rolled by a rolling machine and the leaves are in the form of strips, while broken black tea is rolled and cut by a rolling-cutting machine after withering and the leaves are in the form of granules. Additionally, the rolling and cutting process of broken black tea results in heavier cell fragmentation, larger specific surface areas, and full exposure of the inner components to air, thus accelerating the rate of chemical processes such as oxidation of polyphenols and hydrolysis of glycosides in the tea leaves, and completing the fermentation in a shorter period of time (1 to 2 h) [11]. In contrast, the fermentation time of congou black tea is longer, generally 3 to 6 h [12]. This suggests that fermentation humidity has a much greater effect on congou black tea than on broken black tea, but the relevant information is not yet clear. Against this background, this study intended to optimize the fermentation humidity conditions of congou black tea by setting different fermentation humidity gradients and analyzing their effects on the sensory quality and chemical composition, as well as on the antioxidant activity against DPPH and inhibitory effects on the activities of α-amylase and α-glucosidase, which may offer a theoretical foundation to improve congou black tea’s quality.

## 2. Materials and Methods

### 2.1. Tea Sample Processing

Clonal tea leaves of the “Echa 10” variety (one bud plus two leaves) were picked in late August of 2021 from the South Lake Tea Plantation Base of Huazhong Agricultural University (Wuhan, China). The fresh leaves were mixed well and spread evenly for withering in a 6CWD-200 trough (Zhejiang Green Peak Machinery Co., Ltd., Quzhou, China), with an initial cold blast for 1 h, followed by hot blast withering at 32 °C until 58% moisture content was achieved. After full rewetting, the rolling of leaves was performed on a 6CR-45 rolling machine (Zhejiang Green Peak Machinery Co., Ltd., Quzhou, China) for 2 h in the order of light pressure, heavy pressure and light pressure, followed by taking out the leaves, unblocking them and blending them. Next, the tea leaves were fermented at 32 °C in an RXZ-328A artificial climate chamber (Ningbo Jiangnan Instrument Factory, Ningbo, China) under five different humidity conditions of 55% (RH55), 65% (RH65), 75% (RH75), 85% (RH85) and 95% (RH95), as five treatments in all. Each treatment was performed with three replicates and each replicate contained 1 kg of rolled leaves. Fermentation lasted 200 min, and air circulation was maintained using a circulating ventilation system in the artificial climate chamber. After fermentation, the tea leaves were dried by a 6CTH-6.0 dryer (Zhejiang Green Peak Machinery Co., Ltd., Quzhou, China) with an initial drying at 110 °C for 10 min, then redrying at 90 °C for 30 min and a final drying at 80 °C for 60 min, with sufficient cooling to room temperature and rewetting between each drying step. The resulting dried tea was stored in a −20 °C storage container for subsequent analysis.

### 2.2. Chemical Reagents

Methanol, glacial acetic acid, formic acid and acetonitrile (HPLC) were purchased from Thermo Fisher (Thermo Scientific, Waltham, MA, USA). Theaflavins (standards), catechins (standards), 2, 2-Diphenyl-1-picrylhydrazyl (DPPH, S30629), α-amylase (from Bacilus subtilis, S10003), α-glucosidase (from Saccharomyces cerevisiae, S10050) and 4-nitrophenyl-α-D-glucopyranoside (pNPG, S10137) were purchased from Shanghai Yuanye Biotechnology Co. A Millipore Mill-Q ultrapure water system (Billerica, MA, USA) was used to prepare the purified water (18.2 MX) used in this study. The other reagents and solvents of analytical quality were obtained and used without further treatment, from China National Pharmaceutical Group Corporation (Beijing, China).

### 2.3. Sensory Evaluation

Based on the Chinese National Standard (GB) GB/T 23776-2018, a tea sensory evaluation was performed. Specifically, the congou black tea samples were mixed and placed in different sample trays for appearance evaluation. Next, the tea samples were blind coded by random numbers. After weighing 3 g of dry tea into the evaluation cup, 150 mL of boiling water was added, followed by covering the cup, steeping the tea for 5 min, and then pouring the tea infusion into separate evaluation bowls. At 30 s intervals, five professional tea tasters scored the tea independently on a 100-point scale in terms of dried tea appearance (25%), liquid color (10%), aroma (25%), taste (30%) and infused leaves (10%). The specific sensory features of the black tea were also scored by quantitative descriptive analysis (QDA), including moistness, tightness, tippy hair content, evenness, blending and integrity in appearance; redness and brightness in liquid color; floral and fruity odor; sweetness, grassy and miscellaneous odors in aroma; acidity, bitterness, greenness, astringency, umami, sweetness and concentration in taste; redness, brightness and blending in the infused leaf. The intensity was characterized by a 5-point scale, with 0 for none and 5 for strong.

### 2.4. Chemical Composition Determination

The Folin-Ciocalteu method was used for determining the total amount of polyphenols; the ninhydrin colorimetric method for determining the total free amino acid content; the colorimetric approach involving anthrone and sulfuric acid for determining the amount of soluble sugar; the colorimetric method of aluminium trichloride for measuring the total amount of flavones in tea. Ethyl acetate, n-butanol and ethanol were used for extraction of theaflavins, thearubigins and theabrownines, and the total contents were detected by systematic analysis. The chromatic aberration of the tea infusion was determined with a CS-820N spectrophotometer for color measurement (Hangzhou Caipu Technology Co., Ltd., Hangzhou, China) and the tea infusion was prepared as described for sensory evaluation.

Theaflavin and catechin components were detected by high-performance liquid chromatography (HPLC) (1260 Infinity, Agilent, Santa Clara, CA, USA). Determination of the components of theaflavins was based on GB/T 31740.3-2015 with conditions as follows: column, AgilentTC-C18 (250 mm × 4.6 mm × 5 μm); 35 ℃ column temperature, 0.7 mL/min flow rate, 278 nm detection wavelength, 5 μL injection volume, mobile phase A consisted of 90 mL acetonitrile with 20 mL glacial acetic acid and 2 mL EDTA-2Na (10 mg/mL), with ultrapure water fixed to 1000 mL; mobile phase B consisted of 800 mL acetonitrile with 20 mL glacial acetic acid and 2 mL EDTA-2Na (10 mg/mL). Determination of catechin components was based on GB/T 8313-2018 with conditions as follows: AgilentTC-C18 column (250 mm × 4.6 mm × 5 μm); 35 ℃ column temperature, 0.7 mL/min flow rate, 5 μL injection volume, 278 nm detection wavelength, mobile phase A was ultrapure water (involving 0.1% formic acid), and mobile phase B was methanol (involving 0.1% formic acid). The elution gradient of theaflavins and catechins consisted of 100% A from 0–10 min, 100–68% A from 10–25 min and 68–100% A from 25–35 min.

The volatile compounds were detected by gas chromatography-mass spectrometer (GC-MS), Thermo MS DSQ II (Thermo Fisher Scientific, Waltham, MA, USA). Briefly, aroma components were adsorbed by headspace solid-phase microextraction (HS-SPME), and PDMS/DVB extraction fibers (PDMS/DVB 65 μm) were first aged at 250 °C for 1 h at the GC inlet. Meanwhile, the sample of crushed tea (1.0 g) was placed into a headspace vial (20 mL) and extracted with 5 mL of boiled supersaturated NaCl solution, followed by adding 500 μL of cyclohexanone internal standard solution (0.1 μL/mL), placing the bottle in a 60 °C water bath for 60 min after sealing it promptly and tightly. Finally, SPME was transferred to the GC-MS injection section for the separation, identification and quantification of volatile compounds.

Chromatographic conditions: 30 mm × 0.25 mm × 0.22 μm DB-5MS column; 230 °C inlet temperature; high purity (≥99.99%) helium carrier gas; 1.0 mL/min column flow rate; an initial temperature at 45 °C and held for 2 min, then 7 °C/min to 80°C with no hold, 2 °C/min to 90 °C and held for 2 min, 3 °C/min to 100 °C and held for 2 min, 3 °C/min to 130 °C and held for 2 min, 3 °C/min to 150 °C, and finally 10 °C/min to 230 °C and held for 5 min; 40 °C the column chamber temperature; splitless injection mode. Mass spectrometry conditions: Electron Ionization (EI) ion source; 70 eV electron energy; 230 °C ion source temperature; *m*/*z* 30–500 mass scan range. Electron collision ionization was used for GC-MS analysis. Characterization of volatiles was performed by retention indices (RI, determined from n-alkanes C3-C25) and the mass spectrometry library from the National Institute of Standards and Technology (NIST).

### 2.5. DPPH Radical Scavenging Activity Assay

The scavenging assay of DPPH radicals followed our previously reported method [5]. The concentration gradient of tea extract was 2.0 mg/mL, 1.0 mg/mL, 0.75 mg/mL, 0.5 mg/mL, 0.2 mg/mL, 0.1 mg/mL and 0.01 mg/mL, respectively. Each sample solution (1 mL) was mixed with 1 mL of DPPH (0.15 mmol/L, DPPH was prepared with methanol), followed by incubation of the mixture at room temperature protected from light for 30 min, and then by measuring the 516 nm absorbance of the mixture with a 722N spectrophotometer. The interference of the color of black tea infusion on the experiment outcomes was avoided by setting up a control group with 1 mL of DPPH in the experimental group replaced by 1 mL of distilled water to deduct the background color, and the other operational procedures remained unchanged.

### 2.6. α-Amylase Inhibition Assay

Following Qu’s method [5], 50 μL of tea extract and 50 μL of enzyme solution were quickly mixed, and after a 5-min warm bath at 37 °C, the mixture was supplemented with 1 mL of substrate (2.0 g/L of starch solution). After a 15 min warm bath at 37 °C, the reaction was stopped by adding 1 mL of iodine dilution (0.01 mol/L). Finally, 200 μL of the mixture was suctioned onto an enzyme plate, and a microplate reader was used to measure the absorbance at 660 nm.

### 2.7. α-Glucosidase Inhibition Assay

Based on Qu’s method [5], α-glucosidase solution (100 μL, 1 unit/mL) was prepared in PBS (0.1 mol/L, pH 6.8), quickly mixed with each sample (50 μL), and the 96-well plate was warmed for 10 min at 37 °C. Next, pNPG (50 μL, 2.5 mmol/L) was added to each assay well and warmed for 5 min at 37 °C. Finally, a microplate reader was used to measure the absorbance of each sample at 405 nm.

### 2.8. Statistical Analysis

The results were given as the mean ± standard deviation (SD) (n = 3) with three repetitions of each experiment. Data analysis was performed by one-way analysis of variance (ANOVA). The Fisher’s Least Significant Difference (LSD) approach was used to examine the statistical significance of various treatments, with *p* < 0.05 as significantly different. The data were examined with SPSS Statistics 25.0 program (IBM, Chicago, IL, USA) while Origin 2022 (Stat-Ease Inc., Minneapolis, MN, USA) and Adobe Illustrator 24.0 (Adobe Inc., San Jose, CA, USA) were used to visualize the data. SIMCA 14.1 (Umetrics Inc., Sweden) was used for orthogonal partial least squares discriminant analysis (OPLS-DA). Variable importance in projection (VIP) values were calculated and differential aroma components were screened with the criteria of VIP > 1 and *p* < 0.05. TBtools was used for heatmap analysis of differential aroma components.

## 3. Results and Discussion

### 3.1. Effects of Fermentation Humidity on Congou Black Tea Sensory Quality

Due to different humidity levels during the fermentation process, the five treatments showed differences in the appearance of tea leaves at the end of fermentation (Figure 1). As the fermentation humidity rose, the greenness of the fermented leaves decreased and the redness and gloss rose. Under low fermentation humidity conditions (55%, 65%), the fermented leaves were greenish-yellow and darker, while under high fermentation humidity conditions (85%, 95%), the fermented leaves turned yellowish-red and brighter. Studies have shown that well-fermented black tea leaves will have a bright red color due to the sufficient decomposition of chlorophyll and the generation of thearubigins and theaflavins [13]. In addition, feeling the tea leaves with the hand at the end of fermentation revealed that as fermentation humidity rose, the fermented leaves’ surface became progressively moister. Under low fermentation humidity conditions (55%, 65%), the fermented leaves were drier and had a distinctly prickly feel to the hand, and the trabs were slightly patulous relative to the rolled tea leaves. However, under high fermentation humidity conditions (85%, 95%), the fermented leaves were moister and softer. This is probably due to the increased transpiration and water loss from the fermented leaves at lower relative humidity, resulting in a drier surface [14]. This result suggests that low humidity conditions are unfavorable to congou black tea fermentation.

The sensory assessment results of congou black tea with different fermentation humidity treatments are shown in Table 1, where the tea’s appearance, aroma, taste and infused leaf were all seen to be greatly impacted by fermentation humidity. As shown by the QDA results in Figure 2, as the fermentation humidity increased, the tightness of the dried tea trabs and the shape uniformity gradually improved, coupled with better moistness, agreeing with the change trend in the appearance of the fermented leaves after fermentation. Under high humidity conditions, both the surface and internal leaves in the fermentation baskets were at a similar humidity level, indicative of consistent chemical changes, and no issue of uneven fermentation. In low humidity conditions, the surface-fermented leaves in the fermentation basket lost water more rapidly, causing the fermented leaves to wrinkle, deform and feel prickly, leading to a reduction in the tightness and moistness of the tea leaves after drying. Unlike the surface-fermented leaves, the fermented leaves inside the fermentation basket evaporated water at a slower rate, leading to variations in the degree of deformation and chemical changes in the same basket, hence nonuniform fermentation and a decrease in the evenness of the tea sample. In terms of aroma quality, the tea leaves had a grassy scent for RH55, RH65 and RH75 treatments, with a green tea-like odor in the RH55 and RH65 treatments. Once the fermentation humidity rose to 85%, the grassy scent and miscellaneous odor disappeared and the aroma turned sweet and pure. During black tea fermentation, the content of leaf aldehydes, which present a greenish smell, gradually decreased, in contrast to the content increase of linalool and other aromatic compounds, which have a floral or sweet aroma, thus changing the aroma of the black tea from unpleasantly green to pleasantly sweet, floral or fruity [15]. This indicated that fermentation at low humidity is unfavorable to the formation of black tea’s aroma quality. In terms of taste quality, black teas fermented at less than 75% humidity had a green, astringent and bitter taste, probably due to insufficient fermentation, thus preventing the full oxidation of the tea polyphenols and leading to their high content, especially the retention of the astringent ester-type catechins [16]. In contrast, black teas fermented at 85% humidity or more exhibited a mellow and sweet taste with obvious umami characteristics. This is probably due to the proper retention of polyphenols and their harmonization with other water-soluble substances, thereby enhancing the sensory quality [17]. In terms of tea infusion color, the impact of different fermentation humidity conditions on black tea’s liquid color was relatively small, which was consistent with the chromatic aberration analysis of the tea infusion in Appendix A. Additionally, the high fermentation humidity samples showed better performance in the redness, brightness and blending of infused leaves. The above analysis indicated that 75% fermentation humidity was an important turning point for the sensory quality of congou black tea and a high humidity fermentation environment favored the enhancement of the sensory quality of black tea.

### 3.2. Effects of Fermentation Humidity on Congou Black Tea Taste Composition

The taste of congou black tea is related to the composition and content of its compounds, with tea polyphenols and their amino acids, oxidation products, soluble sugars and flavones as the major components to determine its flavor [2].

In Figure 3A, the content of tea polyphenols was seen to decrease with rising fermentation humidity, exhibiting highly significant differences between the low humidity samples (RH55, RH65) and the high humidity samples (RH85, RH95). A possible explanation is that the high humidity environment could enhance the activity of polyphenol oxidase (PPO) and peroxidase (POD), thus intensifying the enzymatic and other non-enzymatic reactions in the congou black tea, resulting in greater oxidation and lower retention of tea polyphenols [18]. This phenomenon was also observed in tobacco, fruits and vegetables. The higher the relative humidity of the environment, the higher the activity of PPO, and a low humidity environment can cause faster dehydration of the tobacco surface, thus reducing the activity of PPO [19]. Shao et al. found that reactive oxygen species derived from a high humidity environment could stimulate the expression of PPO-related genes in mushrooms, thereby accelerating the oxidation of polyphenolic compounds [20]. Sarpong et al. suggested that a higher relative humidity environment could result in a lower actual temperature than the ambient temperature in dried banana slices, leading to greater retention of PPO and POD activity [21]. In the present study, similar trends were observed for the catechin components and their total amounts (Figure 3I). Specifically, the five treatments exhibited no obvious variations in the composition of simple catechins EGC and C. The ester catechins (EGCG, ECG) and four monomers (EGC, C, EGCG, ECG) of catechins in total showed a significant downtrend with increasing fermentation humidity, with a decrease of 42.5%, 11.8% and 14.5% from RH55 to RH95, respectively. This indicated that the sensitivity to ambient humidity was significantly greater for ester catechins than simple catechins, which is probably associated with the degradation and epimerization of catechins. As previously confirmed, the storage environment of high humidity promoted the degradation and epimerization of catechins in green tea powder [22]. The storage environment at low temperatures and humidity could significantly increase the total flavanols, total polyphenols and ascorbic acid content of green tea [23]. Ester-type catechins have stronger bitterness and astringency than simple catechins, and their higher content in the low humidity treatments may be the main reason for the higher bitterness and astringency of congou black tea [16].

Theaflavins are generated by the oxidative polymerization of catechins, followed by conversion to thearubigins and theabrownins [18]. As indicated by systematic analysis, with the rise of fermentation humidity, the total contents of theaflavins in tea samples gradually decreased, while the total contents of thearubigins and theabrownins gradually increased (Figure 3B–D). Among the theaflavin components determined by HPLC (Figure 3H), there was no significant change in the monomers TF-3′-G and TFDG of theaflavins, while TF, TF-3-G and four monomers (TF, TF-3-G, TF-3′-G, TFDG) of theaflavins in total all showed a significant downtrend. Theaflavins are the main components of the brightness and golden ring appearance of a tea infusion, and also determine the umami and astringency of a tea infusion [24]. Previous studies have revealed thearubigins as the key substances for a tea infusion’s red color as well as its taste concentration and intensity, while showing theabrownins as the main substances for the formation of a tea infusion’s dark brown color and its bland taste [25]. The above results suggest that high humidity fermentation could promote the conversion of theaflavins to thearubigins and theabrownins, thus reducing the astringency of the tea infusion and increasing its redness and intensity.

Amino acids not only contribute to a tea infusion’s umami and sweetness, but also serve as important precursors to black tea aroma formation [26]. In Figure 3E, the content of free amino acids was seen to decrease with the increase of fermentation humidity. These content changes in amino acids are related to their degradation and transformation, and limited protein hydrolysis. The increased oxidation of polyphenols during fermentation at high humidity was reported to facilitate the conversion of amino acids [27]. This phenomenon was also observed in roasting banana chips [28] and in feed production from red clover. When wild-type and low PPO-active red clover were wilted and silaged to measure their free amino acid content during processing, higher levels of free amino acids were found in the low PPO-active red clover than in the normal PPO-active wild-type plants at all stages, and the high PPO-active red clover produced more protein-phenol complexes during silage, thus inhibiting protein hydrolysis and reducing the contents of free amino acids [29].

With increasing fermentation humidity, total soluble sugars (TSS) had an increase of 7.7% in the RH95 treatment versus the RH55 treatment (Figure 3F). As reported by Zuo et al. [30], a storage environment with high humidity and low temperature can help to keep the sucrose, glucose and fructose content of zucchini at a high level. Li et al. compared TSS in straw mushrooms stored at different humidity levels and found that a high relative humidity environment was beneficial for TSS retention [31]. An increase in TSS was also shown to enhance the sweetness and mellowness of a tea infusion [32]. Therefore, high humidity fermentation could significantly improve the sweetness of a tea infusion, thus favoring the formation of quality black tea.

Flavones are a kind of flavonoids, including flavones, flavonols and flavone glycosides. In Figure 3G, their total amount was observed to fall with rising relative humidity, but no significant difference was found between 75% relative humidity and above. Studies have shown that high humidity storage environments could accelerate the decline of flavonoids in Pericarpium Citri Reticulata [33]. The same trend was also observed in the storage of strawberry and tomato peels [34,35]. During black tea fermentation, the total amount of flavonoids was significantly reduced as a result of an enzymatic catalytic redox reaction [36]. This is consistent with in vitro simulations, which showed that when substrates extracted from fresh tea leaves were mixed with distilled water with PPO and POD solutions for fermentation, the presence of PPO could significantly reduce the content of myricetin glucoside in tea, while POD could significantly reduce the content of all flavone glycosides [37]. This indicated that the high humidity environment may accelerate the oxidation process of flavones by increasing enzyme activity. Accordingly, the reduction in the total content of flavones can also alleviate the astringency and bitterness of the tea infusion, thus contributing to tea infusion quality improvement [38]. Correlation analyses were performed between taste sensory scores and compounds (Figure 4). The taste sensory scores were found to have positive correlations with soluble sugars, thearubigins and theabrowns, while significant negative correlations were found with tea polyphenols, EGC, EGCG, ECG, four monomers (EGC, C, EGCG, ECG) of catechins in total, theaflavins, TF, TF-3-G, TF-3′-G, TFDG, four monomers (TF, TF-3-G, TF-3′-G, TFDG) of theaflavins in total and amino acids.

### 3.3. Effects of Fermentation Humidity on Congou Black Tea Volatile Compound Composition

In this study, ninety-eight volatile compounds were identified from the five treatments using HS-SPME-GC-MS, including twenty-seven alcohols, twenty-six esters, twenty-five aldehydes and ketones, seven alkylenes, five acids, four phenols and four other compounds (Appendix A). As shown in Table 2, alcohols were the main aroma substances, accounting for about 59% of the total, followed by aldehydes and ketones at 21%, and the content of phenols was relatively low at only about 1%. The tea aroma varied significantly with different humidity treatments. With the increase in fermentation humidity, the alcohol, alkene hydrocarbon, aldehyde plus ketone and acid contents all gradually increased, in contrast to a content decrease in phenolic substances. The proportion of each substance is shown in Figure 5A, and their proportion was relatively stable. As the fermentation humidity increased, the samples also showed a significant uptrend in the total amount of volatile compounds, with an increase of 6.4% in RH95 relative to RH55. There was a significant difference between low fermentation humidity groups (RH55, RH65) and high fermentation humidity groups (RH85, RH95), with 75% fermentation humidity as the demarcation point.

Based on the aforementioned results, all five treatments could be divided into a high and a low humidity group: group 1 (low humidity) (RH55, RH65) and group 2 (high humidity) (RH75, RH85, RH95). These two groups were investigated by OPLS-DA and the results are shown in Figure 5B. Both R^2^ and Q^2^ values exceeded 0.5, denoting an acceptable fit of the model. After 200 times of permutation tests, the intersection of the Q^2^ regression line with the vertical axis was below zero (Figure 5C), denoting that the model was not over-fitted and validated. The above results showed a good separation effect for the samples fermented under high and low humidity conditions, indicating significant aroma composition variations in congou black tea fermented at different humidity levels. Sixteen differential aroma components under different humidity treatments of congou black tea were screened based on the criteria of VIP >1 and *p* < 0.05 (Figure 5D). As the fermentation humidity increased, a downtrend was observed in the contents of eugenol, 2,4-di-t-butylphenol, furaneol, trans-2-hexenyl hexanoate and nerolidol, while an uptrend was found in the contents of nonoic acid, hexanoic acid, linalool, trans-pyranoid linalool oxide, β-cyclocitral, nerol, α-farnesene and (E,E)-2,4-nonadienal. Meanwhile, the contents of limonene, (E)-2-octenal and jasmone exhibited a trend of increasing first and then decreasing.

Many studies have shown the formation of key aroma substances contributing to black tea odor during the fermentation process [39,40]. In this process, the aroma precursors, including fatty acids (such as linolenic and linoleic), amino acids (such as phenylalanine and methionine), geranyl/farnesyl pyrophosphate and carotenoids (such as β-caroteneand lutein), are correspondingly converted into fatty acid-derived volatiles (FADV), amino acid-derived volatiles (AADVs), volatile terpenoids (VTs) and carotenoid-derived volatiles (CDVs). These volatile compounds make an important contribution to the formation of the characteristic aromas of sweet, floral and fruity odors in congou black tea [3]. For example, linalool and its oxides are the key aromatic substances in black tea, and their content increase is associated with the up-regulated expression of the linalool synthase gene (*CsLIS),* which converts geranyl diphosphate into linalool. Additionally, β-glucosidases can also increase the linalool content by hydrolyzing linalool glycosides [41]. VTs such as nerol, α-farnesene and linalool and its oxides act together as key active compounds to provide floral aromas to black tea [42], while β-cyclocitral, which is attributed to CDVs, provides fruity and faint scents to congou black tea. Based on the available reports, it can be found that substances such as linalool oxides and ionone can increase black tea floral aroma, and benzaldehyde and phenylacetaldehyde can increase its sweet aroma [40]. Combined with Appendix A, all these substances were found to be most abundant in the RH95 treatment, and despite not being recognized as essential fragrance chemicals, they still contributed to the presentation of black tea’s aroma.

Overall, as the relative humidity of the fermentation environment increased, the content of volatile compounds in the black tea increased, leading to a rise in aroma concentration. The contents of linalool, its oxides, nerol and other substances beneficial to black tea aroma quality showed a significant increase, indicating that high relative humidity fermentation is more conducive to the formation of pleasant aromas in congou black tea, which was well supported by the sensory evaluation results.

### 3.4. Effects of Fermentation Humidity on Congou Black Tea Bioactivity

The DPPH radical scavenging rate is usually used to characterize the antioxidant capacity of tea. α-Amylase and α-glucosidase are two essential enzymes involved in carbohydrate digestion, and by inhibiting their activities, blood sugar can be lowered, which is often used in medicine to treat diabetes [43]. The scavenging effect of DPPH free radicals and the inhibitory effects against α-amylase and α-glucosidase activities by black teas fermented under different humidity conditions are shown in Table 3. The half-inhibitory concentrations (IC_50_) indicate the concentration of the sample at 50% scavenging rate or inhibition rate, which is a key indicator for the scavenging or inhibition effect, i.e., a lower IC_50_ value indicted a stronger scavenging or inhibitory effect of the sample. In Table 3, the DPPH scavenging effect and the inhibition of α-amylase and α-glucosidase activities were seen to decline significantly with the rise of fermentation humidity. Under 55–65% humidity conditions, the scavenging effect of tea on DPPH and the inhibition of α-amylase and α-glucosidase activities did not decrease significantly. Under 85–95% humidity conditions, except for a remarkable decrease in the inhibition of α-glucosidase, the DPPH scavenging effect and the inhibition of α-amylase activity also did not decrease significantly. However, there were significant differences between the high relative humidity (85–95%) and the low relative humidity (55–65%) treatments. This indicated that 75% fermentation humidity is an important turning point.

The correlation between the three IC_50_ values and the contents of taste compounds was analyzed for each sample (Figure 6). These three IC_50_ values were all shown to have remarkable negative correlations with polyphenols, catechins (EGCG), theaflavins (TF, TFDG and four monomers of theaflavins in total) and amino acids, while having remarkable positive correlations with thearubigins, indicating that the levels of these substances in black tea can remarkably impact the bioactivity of tea samples. This correlation has been confirmed in storage experiments with white tea [44]. Among these taste components, phenolic compounds such as polyphenols, flavones, catechins and theaflavins are considered as the main and most effective antioxidants against DPPH and inhibitors of enzyme activity. The hydroxyl substituents and aromatic structure of phenolic compounds are the key factors for their antioxidant and free radical scavenging effects [45]. The hydroxyl groups in polyphenols could inhibit the α-amylase activity by binding to the active site of α-amylase, and phenolic compounds containing gallic acyl groups (such as catechins and theaflavins) have a greater inhibitory impact on α-amylase activity due to their ability to provide abundant hydroxyl groups [46]. Similarly, hydroxylation and gallotannoylation of flavonoids in the samples could enhance their inhibitory effects on α-glucosidase activity [47]. He et al. also showed that theaflavins possess a greater ability to provide protons due to their specific structure, resulting in stronger antioxidant activity [48]. Qu et al. found that theaflavins could significantly inhibit α-glucosidase activity, suggesting that elevated levels of theaflavins in tea samples could significantly increase enzyme activity inhibition [5]. Amino acids with phenolic or thiohydroxyl groups (such as tyrosine, cysteine and methionine) have been shown to have stronger antioxidant activities [49]. A large proportion of common amylase inhibitors are proteins mainly derived from plants and streptomyces, and often consist of disulfide bonds and a variety of amino acids [50]. Glutamic acid, arginine and leucine from microalgae have good antioxidant and enzyme activity inhibiting properties due to their small molecular weight, hydrophobic nature or hydroxyl group content [51]. Wang et al. also mentioned that the functional groups of amino acids, such as aromatic and sulfur-containing amino acid residues, were associated with their inhibitory effects [52]. These reports demonstrated a positive correlation of amino acid content with the antioxidant activity and enzyme activity inhibitory effects of tea samples.

The above results indicated that tea samples fermented under low humidity conditions (RH55, RH65) have stronger antioxidant activity and enzyme activity inhibition. However, sensory review results showed that tea fermented at low humidity has a heavier astringent taste and green scent, so its quality is not as good as tea fermented at high humidity. Based on the results of the tea sensory quality review and the bioactivity analysis, an appropriate low humidity fermentation without affecting tea sensory quality is beneficial to enhance the bioactive function of black tea.

## 4. Conclusions

It can be concluded that fermentation humidity mainly affects the appearance, aroma and taste quality of congou black tea (Figure 7). High humidity (85% or above) during fermentation is more conducive to the sensory qualities of congou black tea, with softer, moister fermented leaves, a tighter, drier appearance, a sweet and pure aroma and a stronger taste of sweetness and umami. This can be partly attributed to the oxidation of tea polyphenols, as well as the transformation of catechins, theaflavins, flavones and amino acids, plus the retention of soluble sugars. Additionally, it is also related to the increase of floral and fruity aroma substances, such as linalool, its oxides and nerol. Despite its stronger inhibitory effects on α-amylase and α-glucosidase activities and antioxidant activity against DPPH, the black tea fermented at low humidity (55–75%) showed poorer sensory quality. In summary, the biological activity of tea samples can be maintained by appropriately reducing the fermentation humidity without affecting the tea’s sensory quality, so that drinking this kind of black tea can favor the recovery of the diabetic population. This study offered an effective theoretical foundation for processing high-quality congou black tea under optimal fermentation humidity conditions.

## Figures and Tables

**Figure 1 foods-12-01726-f001:**
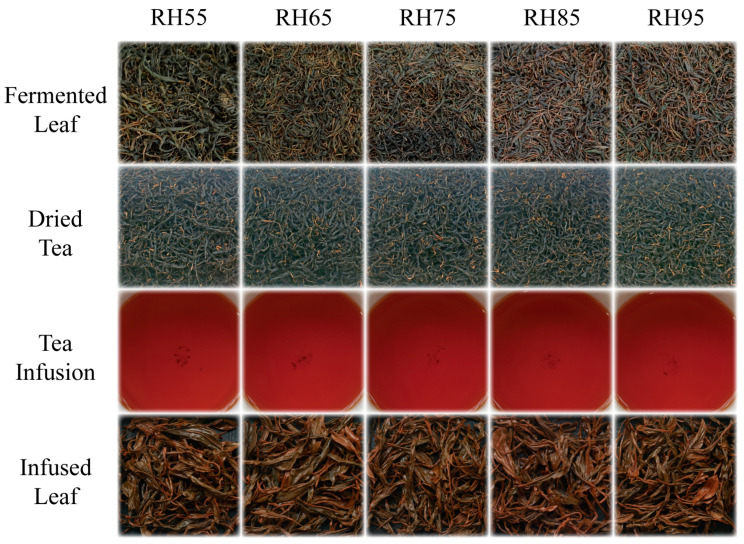
Effect of fermentation humidity on the quality of fermented leaves, dried tea, tea infusion and infused leaf of congou black tea.

**Figure 2 foods-12-01726-f002:**
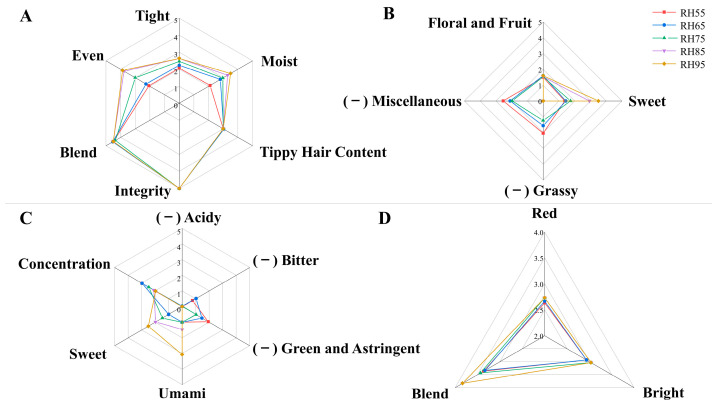
QDA analysis diagrams for the appearance (**A**), aroma (**B**), taste (**C**) and infused leaf (**D**) of congou black tea under different fermentation humidity conditions, with different colored lines for different samples.

**Figure 3 foods-12-01726-f003:**
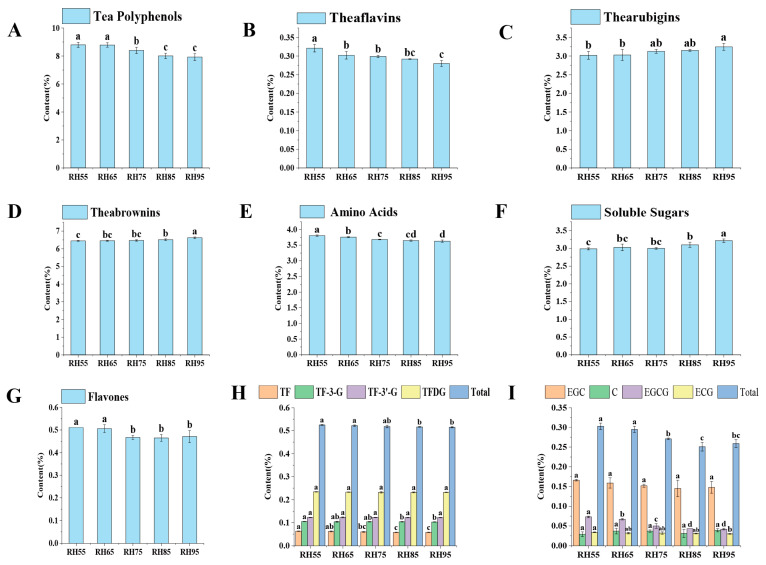
Content of tea polyphenols (**A**), theaflavins (**B**), thearubigins (**C**), theabrownins (**D**), amino acids (**E**), soluble sugars (**F**), flavones (**G**), four main monomers (TF, TF-3-G, TF-3′-G, TFDG) of theaflavins and their total (**H**), and four main monomers (EGC, C, EGCG, ECG) of catechins and their total (**I**) in different treatments. Significant differences at *p* < 0.05 are indicated by different lowercase letters.

**Figure 4 foods-12-01726-f004:**
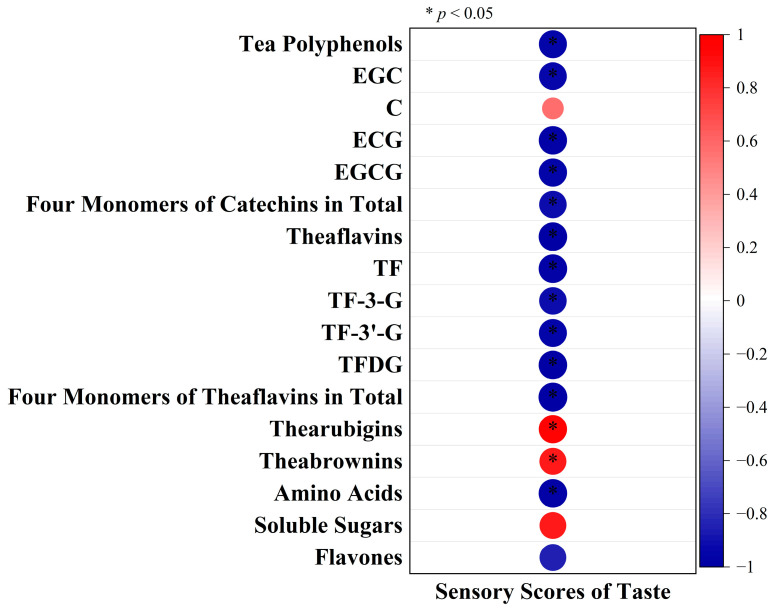
Correlation analyses between taste sensory scores and compounds, with red for positive correlation and blue for negative correlation; darker color and larger circle area indicate larger correlation coefficient; * indicates significant difference at *p* < 0.05.

**Figure 5 foods-12-01726-f005:**
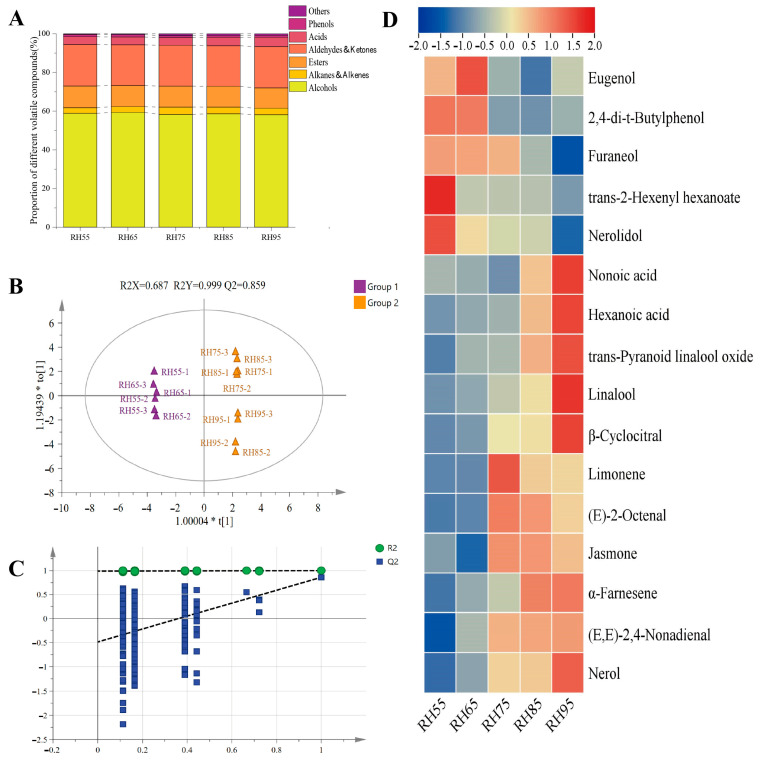
Proportion of different volatile species in the samples (**A**), OPLS-DA score plot of black tea with different humidity treatments (**B**), permutation test of black tea with different humidity treatments (**C**), and cluster heat map of variant aroma components of black tea under different fermentation humidity conditions (**D**).

**Figure 6 foods-12-01726-f006:**
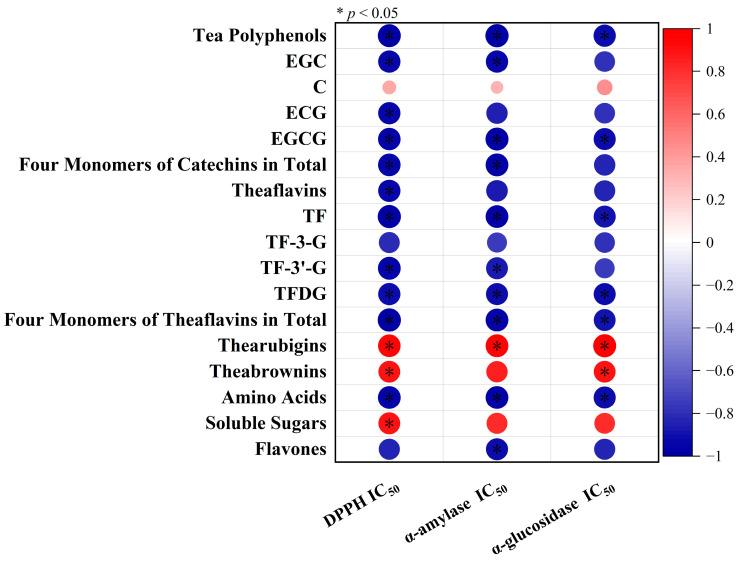
Correlation analysis of IC_50_ values of DPPH, α-amylase and α-glucosidase with taste component contents, with red for positive correlation and blue for negative correlation; darker color and larger circle area indicate a larger correlation coefficient; * indicates significant difference at *p* < 0.05.

**Figure 7 foods-12-01726-f007:**
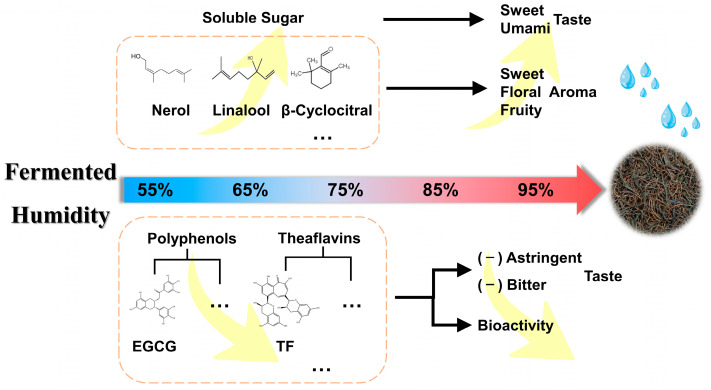
Schematic diagram for the main effects of fermentation humidity on congou black tea quality.

**Table 1 foods-12-01726-t001:** Effect of fermentation humidity on sensory quality of congou black tea.

Samples	Appearance	Infusion Color	Aroma	Taste	Infused Leaf	Total Scores
	(25%)	(10%)	(25%)	(30%)	(10%)	
RH55	78.2 ± 0.2 e	84.8 ± 0.3 a	73.8 ± 0.3 e	72.1 ± 0.1 e	80.1 ± 0.1 bc	76.1 ± 0.1 e
RH65	79.9 ± 0.1 d	84.2 ± 0.3 a	76 ± 0.2 d	77 ± 0.2 d	80 ± 0.2 c	78.5 ± 0 d
RH75	83.8 ± 0.2 c	85 ± 0.9 a	81.2 ± 0.3 c	80.9 ± 0.1 c	80.2 ± 0.1 ab	82.1 ± 0.1 c
RH85	85.1 ± 0.1 b	84.5 ± 0.9 a	83.9 ± 0.1 b	83.9 ± 0.1 b	80.2 ± 0.1 ab	83.9 ± 0.1 b
RH95	86 ± 0.2 a	84.5 ± 0.5 a	87 ± 0.2 a	88 ± 0.2 a	80.4 ± 0.1 a	86.2 ± 0.1 a

Note: Significant differences at *p* < 0.05 are indicated by different lowercase letters in the same column.

**Table 2 foods-12-01726-t002:** Effect of fermentation humidity on the contents of different volatile species in congou black tea. (μg/g).

Substances	RH55	RH65	RH75	RH85	RH95
Alcohols	257.22 ± 5.34 a	258.88 ± 6.44 a	262.55 ± 6.16 a	266.57 ± 16.09 a	270.37 ± 5.33 a
Alkanes & Alkenes	12.64 ± 0.9 b	12.97 ± 0.99 b	16.95 ± 1.54 a	15.79 ± 1.58 a	15.67 ± 0.89 a
Esters	49.21 ± 1.29 a	47.85 ± 1.72 a	48.7 ± 3.28 a	49.07 ± 2.43 a	48.85 ± 2.12 a
Aldehydes & Ketones	93.45 ± 0.68 b	91.6 ± 1.14 b	94.99 ± 3.89 b	95.2 ± 1.47 b	98.99 ± 1.21 a
Acids	17.55 ± 1.45 c	17.73 ± 0.19 c	17.94 ± 1.63 bc	19.91 ± 0.47 b	22.33 ± 1.37 a
Phenols	5.17 ± 0.34 ab	5.35 ± 0.11 a	4.51 ± 0.25 c	4.52 ± 0.19 c	4.83 ± 0.15 bc
Others	1.65 ± 0.07 d	1.87 ± 0.02 c	4.46 ± 0.12 a	3.71 ± 0.07 b	3.81 ± 0.11 b
Total	436.88 ± 5.90 b	436.25 ± 8.52 b	450.11 ± 7.04 ab	454.77 ± 20.71 ab	464.84 ± 6.05 a

Note: Significant differences at *p* < 0.05 are indicated by different lowercase letters in the same row.

**Table 3 foods-12-01726-t003:** DPPH scavenging and inhibition of α-amylase and α-glucosidase activities by congou black tea under different fermentation humidity treatments.

Samples	DPPH	α-Amylase	α-Glucosidase
	IC_50_ (μg/mL)	IC_50_ (mg/mL)	IC_50_ (μg/mL)
RH55	397.27 ± 6.53 c	37 ± 1.17 c	28.63 ± 1.7 c
RH65	404.33 ± 9.87 bc	36.73 ± 1.11 c	27.38 ± 1.79 c
RH75	413.03 ± 3.17 b	47.91 ± 1.55 b	33.65 ± 2.38 b
RH85	431.6 ± 11.25 a	55.19 ± 5.3 a	33.54 ± 1.85 b
RH95	437.07 ± 8.01 a	57.81 ± 5.1 a	38.36 ± 2.17 a

Note: Significant differences at *p* < 0.05 are indicated by different lowercase letters in the same column.

## Data Availability

Data available in a publicly accessible repository. Data is contained within the article or Appendix A.

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
