# Peer review of "Effect of Fermentation Humidity on Quality of Congou Black Tea"

_foods, 2023, doi:10.3390/foods12081726_

Round 1

Reviewer 1 Report

The article presents findings from a study investigating the effect of different levels of fermentation humidity (ranging from 55% to 95%) on the quality and biological activity of congou black tea. The study found that fermentation humidity primarily affected the tea's appearance, aroma, and taste quality. Tea fermented at a low humidity (75% or below) showed a decrease in tightness, evenness, and moistening degree, and had a heavy grassy and greenish scent, and a green, astringent, and bitter taste. In contrast, tea fermented at a high humidity (85% or above) had a sweet and pure aroma, a mellow taste, and an increase in sweetness and umami. As fermentation humidity increased, the content of flavones, tea polyphenols, catechins (EGCG, ECG), and theaflavins (TF, TF-3-G) decreased, while the content of soluble sugars, thearubigins, and theabrownins increased, which contributed to the formation of a sweet and mellow taste. The study also found that the tea's total amount of volatile compounds and the content of alcohols, alkanes, alkenes, aldehydes, ketones, and acids gradually increased as the fermentation humidity increased. The tea fermented at a low humidity had stronger antioxidant activity and inhibitory effects on α-amylase and α-glucosidase activities. Overall, the study suggests that the desirable fermentation humidity for congou black tea should be 85% or above.

Some possible questions that can be asked based on this article are:

Why does fermentation humidity primarily affect the appearance, aroma, and taste quality of congou black tea?

Why does tea fermented at a low humidity show a decrease in tightness, evenness, and moistening degree?

Why does tea fermented at a high humidity present a sweet and pure aroma, and a mellow taste?

Why do the content of flavones, tea polyphenols, catechins, and theaflavins decrease as fermentation humidity increases?

Why is the desirable fermentation humidity for congou black tea suggested to be 85% or above?

Reviewer 2 Report

1.        The efficiency of the fermentation process is affected by the type of bacteria used, the work lacks such information

2.        Check the citations, I see inaccuracies, e.g. line 156 quotes literature [14] that is not related to GC-MS

3.        Figure 1 does not show the tendency of changes in the parameters described in the text under the influence of humidity, they are not visible

4.        Table S1 and others cited in the text is missed- Supplementary Materials

5.        The graphs in Fig.2 are correct in terms of methodology, but readability should be improved

6.        In chapter 3.2, the taste compounds presented are responsible for the appropriate taste and should be presented in such a way

7.        The presented taste compounds in chap. the results should be described in the methodology, no soluble sugars, theaflavin components, catechin components

8.        What is the difference between the theaflavins compounds presented in Fig. 3b and theaflavin-total in Fig. 3h?

9.        The data representation in Fig.3 should be corrected in such a way that the differences between the experiments are visible

10.     Fig. 4a does not show the differences and tendencies presented in the text regarding volatile species in the samples

11.     Used the orthogonal partial least squares discriminant analysis (OPLS-DA) as an analysis method and  the generated heatmap should be described in chapter materials and methods

12.     Interpreting the results on lines 433-436 and concluding that 75% fermentation humidity is an important turning point is incomprehensible without further explanation

13.     I think the main effects of fermentation humidity on the taste, volatile compound composition and antioxidant properties should be summed up in the form of a table or figure

Throughout the work, please pay attention to standardizing terminology, supplementing the presentation of the methods used, compliance of the trends presented in the text and in the figures, improving the readability of the figures, checking cited literature

Round 2

Reviewer 1 Report

The author's response appears to be well-written and grammatically correct. The response provides clear and concise explanations to the questions posed, demonstrating a strong understanding of the topic. Additionally, the response shows the author's ability to synthesize and communicate scientific information effectively. Overall, the response is well-organized, informative, and demonstrates a high level of proficiency in English.